# Circular RNAs: A New Piece in the Colorectal Cancer Puzzle

**DOI:** 10.3390/cancers12092464

**Published:** 2020-08-31

**Authors:** Pinelopi I. Artemaki, Andreas Scorilas, Christos K. Kontos

**Affiliations:** Department of Biochemistry and Molecular Biology, Faculty of Biology, National and Kapodistrian University of Athens, GR-15701 Athens, Greece; partemaki@biol.uoa.gr (P.I.A.); ascorilas@biol.uoa.gr (A.S.)

**Keywords:** circRNA, gastrointestinal cancer, microRNA sponges, transcription regulation, RNA splicing, tumor biomarkers, circularization, peptide translation, therapeutic targets, regulation of carcinogenesis

## Abstract

**Simple Summary:**

Circular RNAs (circRNAs) are a relatively new and unexplored RNA type, implicated in several aspects of cell life, in normal and pathological states. This review aims to discuss the roles of circRNAs in colorectal cancer (CRC) initiation, progression, and therapy resistance, as well as their potential clinical value as CRC biomarkers. CRC is characterized by an elevated mortality rate, poor prognosis of high-grade tumors, high metastatic potential, and resistance to conventional therapies. Therefore, there is an urgent need for identification of novel molecules involved in colorectal carcinogenesis. Prompted by several studies examining circRNA involvement in CRC, in this review we seek to summarize the existing knowledge on circRNA expression in CRC and their implication in cellular pathways and molecular mechanisms underlying colorectal carcinogenesis. Lastly, we discuss the limitations and future perspectives highlighting the missing pieces of the puzzle and aspects of the circRNA research field that should be further investigated.

**Abstract:**

Colorectal cancer (CRC) is the third most fatal type of malignancy, worldwide. Despite the advances accomplished in the elucidation of its molecular base and the existing CRC biomarkers introduced in the clinical practice, additional research is required. Circular RNAs (circRNAs) constitute a new RNA type, formed by back-splicing of primary transcripts. They have been discovered during the 1970s but were characterized as by-products of aberrant splicing. However, the modern high-throughput approaches uncovered their widespread expression; therefore, several questions were raised regarding their potential biological roles. During the last years, great progress has been achieved in the elucidation of their functions: circRNAs can act as microRNA sponges, transcription regulators, and interfere with splicing, as well. Furthermore, they are heavily involved in various human pathological states, including cancer, and could serve as diagnostic and prognostic biomarkers in several diseases. Particularly in CRC, aberrant expression of circRNAs has been observed. More specifically, these molecules either inhibit or promote colorectal carcinogenesis by regulating different molecules and signaling pathways. The present review discusses the characteristics and functions of circRNA, prior to analyzing the multifaceted role of these molecules in CRC and their potential value as biomarkers and therapeutic targets.

## 1. Introduction

Colorectal cancer (CRC) is one of the most well-studied types of human malignancies, due to its high occurrence and mortality rate worldwide. Based on recent surveys, its global burden is estimated to increase by 60%, resulting in more than 2.2 million new cases and 1.1 million deaths by 2030. Its occurrence is higher in the developed countries, but the less developed ones are characterized by a significantly decreased survival rate [1]. The poor prognosis of high-grade tumors, the high potential of metastasis, and the resistance to conventional therapies constitute the greatest challenges in CRC. Additionally, the absence of effective, non-invasive screening tests in the clinical practice hampers the early diagnosis of CRC. Numerous studies have focused on the elucidation of the molecular mechanisms underlying colorectal carcinogenesis, designating the role of several genetic factors, including alterations in chromosomal copy number and stability, microsatellite stability, aberrant gene methylation, and deregulated gene expression [2]. So far, three fundamental gene categories are considered to be implicated in CRC initiation and progression: firstly, tumor suppressor genes such as *APC*, *DCC*, *TP53*, *SMAD2*, *SMAD4*, and *CDKN2A*; secondly, oncogenes such as *KRAS* and *NRAS*, and thirdly, DNA repair genes, including members of the DNA mismatch repair mechanism and *MUTYH* [3]. However, the elevated heterogeneity and the multistep initiation of CRC impede the clarification of this malignancy and the adoption of novel strategies for early diagnosis and successful treatment.

Circular RNAs (circRNAs) are a neglected RNA type deriving from back-splicing. Initially, they were discovered in RNA viroid analysis during the 1970s [4] but were characterized as by-products of alternative splicing. However, the high-throughput analysis revolution uncovered their widespread expression, arousing the scientific interest [5]. circRNAs present the following four particular features: firstly, they form a circular structure, without terminal structures, via a head-to-tail back-splice, and hence show increased stability and resistance to exonuclease decay, compared to the linear RNA molecules [6]. Secondly, their length varies from hundreds to thousands of nucleotides [7] and they have a wide expression pattern [8]. Based on the origin of their sequence, namely intronic or exonic, they are categorized in three subgroups: the exonic circRNAs (EcircRNAs), the circular intronic RNAs (ciRNAs), and the circRNAs with retained intron(s) (EIciRNAs); each category is generated through a distinct circularization mechanism [9]. It is worth mentioning that their mechanisms of biogenesis and the swift of the splicing process from the linear RNA to circRNA have not been fully elucidated yet. Thirdly, their expression pattern has been characterized as cell-type-specific or developmental-stage-specific, but the mechanism accounting for this specificity is also poorly understood [9]. For instance, CDR1as is more highly expressed in murine brain tissues than in non-neural ones [5]. Finally, they are evolutionarily conserved among species [10].

circRNAs have been designated as crucial modulators in several aspects of cell life, both in physiological and pathological states. The current literature supports that circRNAs could be involved in biological processes by acting either as sponges of microRNAs (miRNAs) and RNA-binding proteins (RBPs) [11,12] or by encoding for peptides [13]. Especially, their miRNA-sponging activity, which was first discovered, implies that circRNAs can affect post-transcriptional gene regulation mediated by miRNAs. Additionally, they may modulate their parental gene transcription, and affect cell proliferation and growth [14]. Interestingly, there are studies supporting that the stabilized circRNAs can be retrotranscribed and integrated into the genome, resulting in the circRNA pseudogene formation. For instance, a pseudogene deriving from circ-Rfwd2 (gene origin: *Cop1*) has been observed in the mouse genome [15]. Furthermore, the fact that circRNAs are abundant and stable in exosomes suggests their potential involvement in cell-cell communication mediated through the exosomes [16]. Finally, it has been shown that circRNAs can control epigenetic changes particularly frequent in cancer, such as DNA methylation and histone modifications. circ-FECR1, which has been shown to induce extensive DNA demethylation in the *FLI1* promoter leading to epigenetic activation of this gene, exemplifies this function [17].

Due to the aforementioned features and functions of circRNAs, it has been proposed that they play a pivotal role in the initiation and progression of cancer. Several research studies have even ascribed the typical characteristics of cancer cells to circRNA expression and function. More specifically, in 2000, six hallmarks of cancer, which the progressive conversion of normal cells into malignant ones was attributed to, were proposed. These hallmarks could be categorized as self-sufficiency in growth signals, evasion of antigrowth signals, resistance to cell death, limitless replication potential, sustained angiogenesis, tissue invasion, and metastasis [18]. In 2011, two emerging hallmarks have been added: reprogramming of energy metabolism and evading immune destruction [19]. According to current studies, circRNAs affect all of the aforementioned cancer properties [20,21,22].

Considering all the aforementioned data, circRNAs should be scrutinized in the CRC context. Until now, several studies have focused on their examination, concluding to promising results that underscore the potential role of circRNAs in CRC onset and development. Despite the great progress that has been accomplished so far, many questions remain unanswered and limitations have to be surpassed. This review aims to shed light on the current knowledge regarding the implication of circRNAs in the development of CRC, including proliferation, invasion, metastasis, and treatment resistance, and to present circRNAs which could ideally act as biomarkers and/or therapeutic targets.

## 2. Biogenesis

Three potential models for circRNA biogenesis have been proposed. The mechanism of the biogenesis of most circRNAs is called lariat-driven circularization or exon skipping. According to this mechanism, the pre-mRNA folds partially during transcription, encouraging the attack of the 5’ splicing site (splice donor) of the upstream intron to the 3’ splicing site (splice acceptor) of the downstream intron. This back-splicing of the folded region generates the circRNA, while the remaining exons form a linear mRNA (Figure 1A) [23]. The second model is the intron pairing-driven circularization or the direct back-splicing mechanism. Flanking reverse-complementary sequences at the introns (mostly Alu sequences) mediate back-splicing, generating circRNAs. circRNAs deriving from this procedure can be categorized into two groups: those in which the intron sequence is retained and coexists with the exons (EIcircRNAs) and those in which the intron sequence is removed (EcircRNAs) (Figure 1B) [23]. Merely intronic circRNAs exist, as well; their biogenesis necessitates a consensus motif consisting of a 7-nt GU-rich element near the 5’ splice site and an 11-nt C-rich element near the branchpoint site (Figure 1C) [24].

According to the third proposed model of circRNA biogenesis, RNA-binding proteins (RBPs) play a pivotal role in the generation of some circRNAs. Indicatively, the RBPs bring closer the donor site and the acceptor site via binding the flanks of the introns and hence facilitate circularization (Figure 1D). For instance, muscleblind-like splicing regulator 1 (MBNL1) and Quaking homolog KH domain RNA-binding (QKI) protein are two known RBPs that promote the formation of circ-Mbl and circ-QKI, respectively [7,25,26]. Additionally, it has been shown that the production of several circRNAs is dynamically regulated by the alternative splicing factor QKI, during epithelial-mesenchymal transition (EMT) [25]. Especially, QKI is downregulated in CRC and its overexpression has been linked to the attenuated growth of malignant cells [27]. In CRC, circRNAs exhibit a deregulated expression tendency, as it is analyzed in the following section, which could be attributed to QKI downregulation. However, RBPs can either activate or suppress circRNA generation. For instance, the binding of the RNA-editing enzyme ADAR1 to the double-stranded RNA regions, which promotes A-to-I editing, hinders circRNA biogenesis through promoting the melting of stem structures. Taken into consideration that circular splicing and linear splicing can compete with each other, it is possible that regulation of circRNA biogenesis by ADAR1 acts as a regulatory mechanism for the expression of the linear isoforms [28].

The aforementioned circRNA biogenesis models require a precursor mRNA as a template. However, recent transcriptomic analysis in metazoans has demonstrated that circRNAs may derive from transfer RNAs (tRNAs), as well. More specifically, the intron included in some primary tRNA molecules can generate a tRNA intronic circular RNA (tricRNA) (Figure 1E), but the exact mechanism of biosynthesis and the potential cellular function of tricRNAs warrant further elucidation [29].

Additionally, a model of alternative circularization suggests that more than one circRNAs and linear RNAs can be generated from a single gene, via RNA-pairing competition. More specifically, complementary sequences within distinct introns favor linear mRNA generation, while complementary sequences in different introns promote circularization. The competition between these reverse complementary sequences can lead to the biogenesis of multiple alternative circRNAs (Figure 2). However, alternative circularization is a rather complicated process, in which RBPs are implicated [30]. Differential distribution of the complementary sequences among species suggests that alternative circularization could be species-specific.

## 3. Functions in CRC

Functional analysis of circRNAs has highlighted their potential miRNA-binding sites, suggesting their implication in CRC. miRNAs are significant post-transcriptional regulators; deregulation of their expression levels is associated with CRC pathogenesis. Several studies support that some non-coding RNAs possess miRNA-response elements (MREs) and can act as competing endogenous elements (ceRNAs). More specifically, they compete with mRNAs for the binding of miRNAs, thus attenuating the regulatory effects of miRNAs on their targets [31]. Through their miRNA-sponging action (Figure 3A), several circRNAs regulate the proliferation and migration of CRC cells and affect therapy resistance. For instance, circ-CDYL has been shown to bind to miR-150-5p, the upregulation of which has been associated with the downregulation of PTEN and phosphorylation of PI3K, AKT, JAK2, and STAT5, which are key players in CRC signaling [32].

Another intriguing function of circRNAs in CRC is mediated through their direct interaction with proteins. The nuclear translocation of transcription factors, the cytoplasmic translocation of proteins, the regulation of function and stability of proteins, the promotion or suppression of protein-protein and protein-DNA interactions constitute the major functions of circRNAs as protein-binding partners [33]. More specifically, in CRC, circ-ACC1 has been suggested to form a ternary complex with the regulatory β and γ subunits of PRKAA1 (AMPK), stabilizing it and increasing its activation, thereby promoting fatty acid β-oxidation, glycolysis, and growth of CRC cells (Figure 3B) [34]. In breast cancer, circ-FLI1 is thought to directly recruit TET1 demethylase to the *FLI1* promoter, resulting in DNA demethylation. Moreover, circ-FLI1 is also considered to bind to the promoter of *DNMT1* gene and reduce the expression levels of this DNA methyltransferase. The combinatorial effect of the two aforementioned actions of circ-FLI1 is the elevated expression of FLI1 protein. Therefore, circ-FLI1 is thought to play a role in regulating breast tumor development, through the FLI1-stimulated promotion of the proliferation of malignant breast cells [17]. As epigenetic alterations are crucial in CRC initiation and progression, it would be interesting to investigate circRNAs interacting with other components of the epigenetic machinery. Additionally, the localization of circRNAs can be either cytoplasmic or nuclear. For instance, EIcircRNAs such as circ-EIF3J and circ-PAIP2, have been reported in the nucleus. Their capability to recruit RNA polymerase II and U1 small nuclear ribonucleoprotein advocates their role in the activation of gene transcription [12]. To the best of our knowledge, similar interactions have not been examined in CRC and therefore, the deregulation of gene expression levels along with the role of circRNAs as transcriptional modulators could be scrutinized, adding one branch in CRC pathogenesis. However, every conclusion should be carefully drawn, since the circRNA-protein interactions are dynamic due to the innate dynamic expression of circRNAs, which allows the adoption of distinct tertiary structures and the maintenance of diverse spatial and temporal expression patterns.

Initially, it was believed that circRNAs are a non-coding RNA type, because they bear neither a 5’-cap structure nor a 3’-poly(A) tail, while they also lack a typical internal ribosome entry site (IRES). Recent data have changed this notion, as a cap-independent manner of circRNA translation was revealed [35]. Legnini et al. discovered circ-ZNF609, which is abundant in polysome fractions and encodes for a protein through a mechanism dependent on splicing but not on 5’-cap presence [36]. Additionally, it has been suggested that consensus N^6^-methyladenosine (m^6^A) motifs are abundant in circRNAs and a single m^6^A site is sufficient to drive translation initiation. This process requires the initiation factor eIF4G2 and the m^6^A reader YTHDF3 [37]. These data support the idea that circRNAs can be translated despite the lack of a 5’-cap structure. Translatable circRNAs have been reported in CRC, as well. Indicatively, circ-LGR4, a circRNA upregulated in CRC, can encode for a peptide with the following amino acid sequence: LQTASDESYKDPTNIQLSK. This peptide has been shown to activate Wnt/β-catenin signaling through its interaction with LGR4, resulting in increased CRC stem cell renewal, carcinogenesis, and invasion (Figure 3C) [38]. Another interesting study proved that circ-PPP1R12A encodes for a functional protein, named circ-PPP1R12A-73aa. This peptide is likely to exert a stimulating effect on the Hippo-YAP signaling pathway both in vitro and in vivo, and therefore to promote proliferation and metastasis [39]. Even though little is known about translatable circRNAs, a web tool predicting the coding ability of circRNAs, Circ-Code, with increased sensitivity and decreased false-positive rate, has been designed [40]. Overall, the translatable circRNAs are expected to be the subject of numerous future research studies.

## 4. Expression of circRNAs in CRC

Several high-throughput experiments suggest a deregulated circRNA expression profile in CRC. Chen et al. have characterized 10,245 differentially expressed circRNAs in CRC tissues, compared to adjacent non-cancerous ones; 6264 circRNAs were found to be upregulated, while 3981 were downregulated [41]. Bachmayr-Heyda et al. have also identified 11 upregulated and 28 downregulated circRNAs in CRC, through differential gene expression analysis of normal colon mucosa and paired cancerous tissues [42]. Their RNA-seq data showed that the ratio of circRNAs to their respective linear transcripts is often decreased in CRC samples and cell lines, compared to non-cancerous ones. This phenomenon seems to be independent of the expression tendencies of single circRNAs and linear transcripts, while the aforementioned ratio and the cell proliferation index are inversely associated [42]. The same tendency has also been observed in other research studies [43,44]. Interestingly, following the comparison of circRNA expression profiles among CRC cell lines with distinct *KRAS* mutational status, Dou et al. demonstrated that circRNA expression levels in *KRAS*-mutated cells are significantly lower than in the *KRAS*-wild type cells, while they showed that circRNAs can be transferred to exosomes, where they are abundant [45]. Finally, an investigation of circRNA expression in metastatic CRC cell lines uncovered 623 differentially expressed circRNAs between metastatic and non-metastatic state, suggesting the participation of circRNAs in CRC development and metastasis [46].

## 5. Proliferation and Progression

circRNAs are, also, implicated in CRC proliferation and progression, acting not only as tumor-suppressors but also as oncogenes. One of the initially identified circRNAs was CDR1as (also known as ciRS-7). Its upregulation has been reported in several malignancies, including CRC, while it has also been correlated with poor OS and increased CRC cell proliferation. A potential mode of action of CDR1as is via binding to and sequestering miR-7. Interestingly, several binding sites for miR-7 have been observed in CDR1as. miR-7 is considered as a tumor suppressor, due to its ability to inhibit several signaling pathways via downregulating *EGFR*, *IGF1R*, and *PAK1* expression. Considering the fact that EGFR mediates cell proliferation, apoptosis, and differentiation, and given the positive association between CDR1as and both *EGFR* and *IGF1R* expression, CDR1as has been suggested to act through the miR-7/*EGFR* and miR-7/*IGF1R* axes [47]. 

Interestingly, a miR-7–independent mechanism of action for CDR1as has been disclosed. More specifically, CDR1as overexpression has been shown to result in the upregulation of PD-L1, CMTM6, and CMTM4, with the two latter having recently been identified as regulators of PD-L1 stability. PD-L1 inhibits T-cell activation, while its overexpression has been correlated with unfavorable prognosis in CRC patients [48]. *CMTM4* and *CMTM6* mRNAs do not seem to possess binding sites for miR-7, suggesting that their regulation by CDR1as could be mediated through the modulation of the function and expression of their respective transcription factors. The identification of a potential linkage between circRNA transcriptomics and immune checkpoints is pioneering, while it provides a deeper understanding of the molecular basis of CRC and suggests a multitherapy approach. Undoubtedly, this mechanism needs further investigation, but these findings could be considered as the first steps towards the clarification of the multifaceted role of circRNAs [49].

Another newly discovered circRNA with oncogenic function in CRC is circ-DENND4C. This circRNA acts as a sponge for miR-760, thus affecting SLC2A1 expression and glycolysis. Since increased glycolysis is essential for cancer cell proliferation, this circRNA could give an insight into CRC pathobiology and the aspect of increased proliferation rate, in particular [50]. Another circRNA with remarkable function is circ-ITCH, which acts as a tumor-suppressor via increasing the expression of its linear *ITCH* transcripts by its miRNA-sponging activity. The upregulation of *ITCH* leads to the ubiquitination and degradation of phosphorylated DVL2 and, thus, to inhibition of the Wnt/β-catenin signaling pathway [51]. Furthermore, circ_0021977 (gene origin: *PSMC3*) is downregulated in CRC tissues and cells, and seems to exert its action via the miR-10b-5p/CDKN1A and miR-10b-5p/TP53 regulatory axis. Due to the downregulation of this circRNA, miR-10b-5p is free to bind to CDKN1A and TP53, and hence decrease their expression and ability to regulate cell proliferation and differentiation.

Additionally, circRNAs could affect the epigenetic status of genes via the regulation of important epigenetic players, such as EZH2. This methyltransferase promotes carcinogenesis in several malignancies by altering the expression of numerous tumor suppressor genes [52] and DNA repair, probably via regulation of RPA3 [53], which binds and stabilizes the single-stranded DNA intermediates generated during DNA replication or upon DNA stress. They could, also, have an impact on the transcription of key genes for cellular viability through modulation of the expression of transcription factors, including the transcription factor AP-1, IRF4, and CDX2 [54]. The latter has, also, been used as a biomarker in diagnostic surgical pathology for gastrointestinal and, especially, colorectal differentiation [55]. The function of the aforementioned circRNAs has an impact on CRC cell proliferation. A more extended list of circRNAs implicated in CRC cell proliferation is presented in Table 1.

## 6. Metastasis and Invasion

Several circRNAs have been investigated in the context of their potential implication in the suppression or promotion of metastasis in CRC (Table 1). However, the mechanism through which circRNAs modulate metastasis remains unclarified. For instance, decreased expression of circ-SMAD7 in CRC tissues compared to adjacent normal ones has been detected and is associated with unfavorable overall survival; its increase results in inhibition of cell migration and invasion. Additionally, it has been shown that the EMT in CRC is suppressed via the overexpression of circ-SMAD7 [82].

Interestingly, the circRNA expression profile has been associated with distant metastasis, and particularly lung metastasis, which is considered as the most common extra-abdominal site of metastasis in CRC. After the comparison of the circRNA expression profiles between CRC tissues from patients with lung metastasis and those from patients with non-metastatic CRC, 431 differentially expressed circRNAs were detected; 192 were found upregulated, while 239 were downregulated. Computational analysis revealed that the genes with upregulated expression of their circRNAs participate in DNA repair, while the genes generating the downregulated ones are involved in signal transduction. Intriguingly, the downregulated circRNAs were found to be involved in the NFKB1 and Wnt/β-catenin signaling pathways in the CRC tissues deriving from patients with lung metastasis. In the same study, three additional circRNAs (circRNA_105055, circRNA_086376, and circRNA_102761) sequestering miR-7 and, consequently, affecting *PRKCB*, *EPHA3*, *BRCA1,* and *ABCC1* expression, were identified [88]. RNA-seq profiling of circRNAs in liver metastasis of CRC revealed a differential expression pattern between tissue samples from CRC patients with and without liver metastasis, while circRNA_0001178 and circRNA_0000826 emerged as potential diagnostic indicators of liver metastases [89].

Moreover, data from a recent research study revealed a potential function of nuclearly localized circ-LONP2 as a key metastasis-initiating molecule and novel co-factor in the microprocessor complex. This circRNA probably interacts with DDX1 and modulates pri-miR-17 processing. Besides its role in the intracellular maturation of miR-17, circ-LONP2 may be implicated in the intercellular transfer of this miRNA, resulting in the spread of metastasis-initiating ability to CRC cells with low metastatic potential and, finally, in the establishment of distant metastasis [68]. To the best of our knowledge, circ-LONP2 is the first circRNA identified as an accessory component of the microprocessor complex, expanding the research on potential regulatory functions of circRNAs in both physiological and pathological processes.

Interestingly, several circRNAs exert their function as promoters or inhibitors of metastasis via modulating key signaling pathways in CRC, including Wnt/β-catenin [90], Ras [79], and Hippo-YAP [39]. Finally, some circRNAs seem to interact with matrix metallopeptidases, particularly with MMP2 [69] and MMP14 [77]. These enzymes are essential for extracellular matrix degradation, thus promoting EMT and metastasis. Especially, MMP2 overexpression has been associated with an unfavorable outcome for CRC patients [91].

## 7. Resistance to Therapy

The main weapons against CRC diagnosed at an advanced stage are radiotherapy, chemotherapy, and targeted therapy. 5-fluorouracil (5-FU) is the standard treatment of CRC, while multi-agent regimens, such as FOLFOX (5-FU and oxaliplatin) and FOLFIRI (5-FU and irinotecan), are administered as well. Targeted therapy has flourished in the last 20 years; novel targets have emerged and several additional therapeutic agents have entered clinical trials. Despite the great progress in this field, resistance to therapy is a major obstacle in effective CRC treatment. Therapy resistance is divided into primary and acquired; so far, it has been attributed to several deregulated genes and signaling pathways. However, further relevant research is essential, since many questions remain unanswered.

Recent studies have revealed that the expression of particular circRNAs affects tumor sensitivity to chemo-and radiotherapy. Indicatively, the knockdown of circ_0001313, which is upregulated in radioresistant CRC tissues compared to radiosensitive ones, induces radiosensitivity by promoting apoptosis, through sequestering miR-338-3p and subsequently increasing CASP3 activity. Therefore, this circRNA constitutes a potential tumor biomarker of radioresistance and, simultaneously, a promising therapeutic target to overcome radioresistance in CRC [61]. A comparative microarray analysis between 5-FU–resistant CRC cells and parental chemosensitive cells has revealed 71 differentially expressed circRNAs, three of which (circ_0007031, circ_0000504, and circ_0007006) have been further analyzed and suggested as significant predictors of chemoradiation resistance in CRC [92]. Interestingly, circ-DDX17 was downregulated in CRC tissues and its high expression was positively associated with chemosensitivity, through its potential action as a sponge for miR-31-5p. This miRNA binds to *KANK1* mRNA and inhibits its expression. KANK1 overexpression has been correlated with restrained tumor invasion and growth in lung and gastric cancer, while its overexpression in glioma seems to lead to curbed tumor growth via enhancing apoptosis [93]. Moreover, circRNAs were investigated in the context of FOLFOX therapy resistance, uncovering a wide range of differentially expressed circRNAs in resected tumors from responders and non-responders to this type of treatment [94].

Despite these promising results, it is essential that circRNA expression profiling and functional analysis be conducted in the context of resistance to other types of CRC treatment, including targeted therapy. For instance, the anti-EGFR therapy resistance arouses researchers’ interest and remains unexplored, concerning the circRNA function and involvement. In Table 2, prominent examples of circRNAs related to resistance to therapy are mentioned. These findings suggest that circRNAs can potentially be exploited as targets for drug sensitization.

## 8. Biomarkers 

According to the definition of the National Cancer Institute, a biomarker is “a biological molecule found in blood, other body fluids, or tissues, characteristic of a normal or abnormal process, or of a condition or disease”. circRNAs have been proposed as biomarkers for CRC diagnosis and prognosis, due to their deregulated expression in CRC tissues compared to their adjacent, normal counterparts, as well as due to their increased stability and half-life. For instance, elevated expression of CDR1as is associated with tumor size, TNM stage, and poor overall survival of CRC patients, indicating its potential value as a prognostic biomarker [65]. Additionally, a recent study proposed a panel of three circulating circRNAs in plasma, with independent diagnostic value regarding carcinoembryonic antigen (CEA)-negative and CA19-9–negative CRC. More specifically, the levels of circulating circ-CCDC66, circ-ABCC1, and circ-STIL were significantly lower in the plasma of CRC patients, compared to healthy controls. circ-CCDC66 and circ-ABCC1 levels were also downregulated in precursor lesions of CRC and were able to diagnose early-stage CRC.

Moreover, circ-ABCC1 expression was inversely associated with tumor growth and disease progression [96]. Furthermore, the levels of circ_0082182 (gene origin: *FAM71F2*), circ_0000370 (gene origin: *FLI1*), and circ_0035445 (gene origin: *ALDH1A2*) in tissues of CRC patients are deregulated, compared to healthy individuals; their combined assessment has led to a molecular signature with high diagnostic accuracy. An additional research revealed the value of a four-circRNA signature [hsa_circ_0122319 (gene origin: *PLOD2*), hsa_circ_0087391 (gene origin: *AGTPBP1*), hsa_circ_0079480 (gene origin: *ISPO*), and hsa_circ_0008039 (gene origin: *PRKAR1B*)] regarding the prediction of postoperative disease recurrence in patients with stage II/III colon cancer [97]. More circRNAs with prognostic and/or diagnostic value are presented in Table 3.

Despite the unique features of circRNAs rendering them ideal biomarkers, most of them are not sensitive or specific enough to be applied in the clinical routine. Therefore, one should be very careful regarding the conclusions deduced and carefully investigate the expression of circRNAs both cellularly and extracellularly, prior to concluding to their prognostic and/or diagnostic value. Additionally, clinical surveys must be conducted in large and diverse groups of patients with long-term follow up information for the validation of the existing results.

## 9. Therapeutic Targets

circRNAs are implicated in several aspects of cancer cell life, as well as in CRC initiation and progression. Even though their biological function needs further elucidation, several of them could be exploited as effective therapeutic targets and/or agents for CRC. So far, there are no preclinical data on targeting or delivering circRNAs as a cancer treatment strategy have been reported; however, two potential ways regarding the future therapeutic use of circRNAs have been proposed; firstly, the regulation of endogenous disease-linked circRNAs either through therapeutic knockdown or ectopic expression, and secondly, the in vitro synthesis of circRNAs with specific properties, e.g., miRNA-sponging activity. Nonetheless, both approaches have major limitations; the ectopic expression or the knockdown of specific circRNAs could lead to off-target effects and the delivery of the in-vitro engineered circRNAs is not efficient. More specifically, circRNAs are not sufficiently hydrophobic to perfuse the cell membrane’s phospholipid bilayer. Additionally, the in vitro synthesis of circRNAs is not cost-effective when performed at a high-scale, as it requires substantial amounts of recombinant RNA ligases. Therefore, alternative approaches have recently been proposed, including allosterically regulated ribozymes. These can successfully mediate circularization, yet a co-factor is also required for increased circRNA stability. Hybrid strategies have been proposed, as well, where the high-scale production of circRNAs takes place in yeasts—a process which is easily subjected to genetic control. Taking into consideration that the technological insight of the strategies adopted for surpassing the main obstacles in the usage of linear RNAs as therapeutic means, could be, in some cases, directly transferred to circRNA-based therapeutic approaches, one could be optimistic regarding the integration of circRNAs in clinical trials as effective anticancer treatment strategies [100].

## 10. Exosomes

Exosomes are extracellular vesicles containing multiple proteins, lipids, DNA, and different RNA types, including circRNAs, while they are considered essential for cell-cell communication. Several findings support the implication of exosomes in the initiation and progression of malignancies. An interesting study revealed that circRNA concentration in exosomes deriving from liver cancer cells was higher than in cells. Surprisingly, the correlation between the exosomal circRNAs and the cellular circRNAs was not strong, implying an actively regulated sorting procedure determining which circRNAs are destined for the exosomes. Even though the actual sorting mechanism of circRNAs to exosomes is unknown, it has been proposed that their transportation to exosomes could take place through the binding of circRNAs to RBPs [101]. Additionally, it was shown that the incorporation of circRNAs in exosomes was higher than the one of the linear RNAs. The ratio of circRNAs to their linear counterparts in exosomes is estimated to be approximately 6-fold higher than in cells. Based on the abundance of circRNAs in exosomes, these molecules were investigated in the blood of CRC patients compared to healthy donors; thus, a differential expression pattern of circRNAs emerged, underlining their potential usage as CRC biomarkers [16].

A recent study investigated circRNAs in serum exosomes of CRC patients and concluded that exosomal circ_0004771 (gene origin: *NRIP1*) is upregulated in the serum of CRC patients, compared to healthy controls; however, the levels of this circRNA were lower in CRC cell lines and tissues, compared to normal colorectal mucosa [102]. This difference is in consistency with the conclusions of the former study and could be explained by the notion that the active transport of circRNAs to exosomes might constitute a mechanism for circRNA clearance [103]. Due to their abundance in exosomes isolated from body fluids, their introduction in clinical practice as diagnostic biomarkers could prove to be beneficial, especially for CRC patients, as the lack of non-invasive biomarkers of early diagnosis is still one of the greatest challenges for researchers [104].

Recent studies have also examined the potential role of exosomal circRNAs in chemoresistance and have revealed that exosomes, as mediators of intercellular signal transduction, deliver the circRNAs from drug-resistant to drug-sensitive cells [95]. Therefore, exosomal circRNAs possibly have a functional role in cancer cell biology.

The biogenesis of exosomes is an interesting and not completely understood research topic. The RAB family of proteins and especially RAB11, RAB27, and RAB35 are implicated in this process [105]. A recent study has shown that circ_0000218 (gene origin: *DCLRE1C*) enhances CRC cell proliferation via the miR-139-3p/RAB1A axis. RAB1A is a small GTP enzyme participating in vesicle transport from the endoplasmic reticulum to the Golgi apparatus, in cell migration and autophagy regulation [59]. As there are experimental indications of regulation of RAB proteins by circRNAs, a future investigation of the modulation of more RAB proteins by circRNAs and the impact of this interaction on exosome biogenesis and, consequently, cell-cell communication would be appealing.

## 11. Bioinformatic Tools

So far, several bioinformatic tools have been designed for the identification of circRNAs in high-throughput sequencing experiments, including CIRI, find_circ, CIRCexplorer, KNIFE, MapSplice, and circRNA_finder. The annotation of circRNAs with these tools is based on the identification of the back-splice junction, which distinguishes circRNAs from linear RNAs. For the identification based on the back-splice junction, either segmented reads or a pseudo-reference can be used. The first approach is based on splitting the sequencing reads, whereas the latter is based on a pre-defined back-splice junction and its flanking sequences in a circRNA; next, the sequencing read is directly mapped against this pseudo-reference for identification of a back-splice site [106].

However, prior to identification, library construction and sequencing are required. Several library preparation protocols can be implemented. To obtain optimal results, it is recommended that circRNA enrichment is performed prior to circRNA library construction. This can be achieved via treatment of total RNA with RNase R, an exoribonuclease which degrades linear RNA molecules, poly(A) depletion, rRNA depletion, or combination of the aforementioned strategies. circRNA sequencing reads can be either single-end or paired-end [106].

## 12. Limitations and Challenges

The field of circRNA research is still in its infancy and several challenges need to be confronted. Firstly, the diversity in nomenclature creates ambiguity and a standard naming system should be established. Even circBase, which is considered as the most updated database [107], has remained static. For this obstacle to be countervailed, the establishment of the prefix “circ-” followed by a 7-digit number, like the circRNA ID in the circBase, for the nomenclature of every circRNA in the literature could be a solution to this problem. Furthermore, the gene origin of each circRNA must be reported.

Secondly, experimental limitations regarding identification, quantification, and validation exist. The peculiarity of the back-splice junction in circRNAs, their sequence similarity with linear transcripts as well as their lower abundance, in some circumstances, compared to their linear counterparts render circRNA determination challenging. High-throughput RNA sequencing is used for circRNA detection as it is a relatively unbiased procedure; however, future approaches could also adopt novel technologies for circRNA detection, such as long-read third-generation RNA sequencing, which gives information about the entire exon structure and not only the back-splice junction. Furthermore, there is a lack of a gold-standard tool for data comparison and the existing bioinformatic algorithms use different strategies for circRNA detection [108]. Therefore, for the reduction of false-positive results, a combination of different algorithms should be exploited. Furthermore, the validation of results remains a hurdle. In the vast majority of the experiments, RT-PCR has been exploited, but the conclusions drawn should be carefully examined, as the template-switching during reverse transcription and exon concatamers generated during the amplification of cDNA molecules deriving from linear mRNAs may lead to false-positive or biased results [109]. Therefore, Northern blotting is a gold-standard alternative for the validation of novel circRNAs; RT-PCR is not required [109]. Another interesting alternative is the in situ hybridization, which can provide spatial information and distinguish circRNA expression patterns between cancer and non-malignant cells within the tumor [109].

Thirdly, despite the extensive research in the field of circRNA biogenesis, the knowledge of the fate and metabolism of circRNAs in cells is still limited. Given the increased stability of circRNAs, their accumulation could be toxic for cells. To date, some hypotheses regarding the export of circRNAs from cells have been stated; the transport to exosomes is the most predominant one. However, little is known about the factors which regulate the circularization, metabolism, and turnover of circRNAs.

Fourthly, it is important mechanistic insights into the biological functions of circRNAs to be gained. So far, the mechanism exploration is conducted via circRNA overexpression or knockdown experiments. However, the off-target effects and the collateral impact on the linear counterpart are great challenges. Alternatively, an optimized CRISPR-Cas system would offer a more robust molecular platform to carry out these studies. Additionally, many studies focus on the function of circRNAs as miRNA sponges, ignoring the rest potential functions of circRNAs. For the characterization of circRNAs as miRNA sponges, the stoichiometric relation between these two molecules and the number of target sites of the circRNA should be taken into consideration [110]. A summary of the existing challenges is illustrated in Figure 4.

## 13. Future Perspectives of circRNAs in CRC

Global genomic hypomethylation, gene promoter hypermethylation, histone modifications, and alterations of miRNA expression patterns are major epigenetic changes in CRC, while largely different alternative splicing events are rather common in this human malignancy [111]. Therefore, it would be interesting to investigate whether the aberrant expression of circRNAs observed in CRC could be, at least partly, attributed to epigenetic changes in the genomic locus from which these circRNAs are produced. Additionally, circRNA expression in CRC may be affected by mutations in *cis*-regulatory elements, including inverted repeats and protein-binding DNA regions; therefore, their role in circRNA biogenesis deserves further investigation.

An additional, largely unexplored perspective of circRNA biology is their susceptibility to post-transcriptional modifications. Chemical modifications of RNA are significant for the regulation of their coding or non-coding activity and stability, while many factors responsible for these modifications are mutated or aberrantly expressed in CRC [112]. Therefore, the biogenesis and function of circRNAs could be inhibited or promoted in CRC, provided that these RNA molecules are subjected to extensive modifications.

It is common knowledge that CRC results from cumulative alterations of the genome and cell properties. Moreover, the characterization of the tumor and the development of a successful treatment strategy entail the assessment of several cancer-related molecules, including immune system components. Therefore, future studies should examine circRNA expression and function in a wider context, in which other molecular changes are co-evaluated, rather than individually. To the best of our knowledge, except for one study which links circRNA expression with *KRAS* mutations, the majority of research studies do not examine such potential relations [45].

Chromosomal translocations are a distinctive characteristic of carcinogenesis and fusion genes are considered as promising molecular targets for cancer therapy. Recent studies highlight the role of fused genes in CRC development, even though this research topic has not been thoroughly investigated [113]. Indicatively, it has been shown that *NAGLU*-*IKZF3* and *RNF121*-*FOLR2* have carcinogenic effects in CRC and could act as novel molecular targets for tailored therapies [113]. In addition to fused genes, fusion circRNAs (f-circRNAs) with potential carcinogenic activity have been reported in leukemia. They can facilitate the malignant transformation of normal cells and confer treatment resistance, usually by integration with other cancer-promoting signals. Thus, f-circRNAs may constitute important targets for antileukemic drugs [114]. Therefore, the investigation of f-circRNAs in CRC would probably provide new insights into the understanding of CRC initiation and progression.

## 14. Conclusions

Overall, circRNA biogenesis derives from back-splicing, but the regulation and the frequency of this event are under investigation, while many functions have been attributed to this RNA type, including miRNA sponging, RBP sponging and/or scaffolding, and peptide translation. Due to their unique characteristics, their participation in tumor development, invasion, and metastasis composes a hot topic in CRC research. Additionally, their potential value as prognostic and/or diagnostic biomarkers has just emerged; their abundance and stable expression in exosomes render circRNAs appealing as candidate biomarkers for non-invasive diagnosis. Even though the multifaceted role and the involvement of circRNAs in colorectal carcinogenesis have just begun to unravel and many questions remain unanswered, the existing data regarding the establishment of circRNAs as potential biomarkers and targets in CRC are quite encouraging.

## Figures and Tables

**Figure 1 cancers-12-02464-f001:**
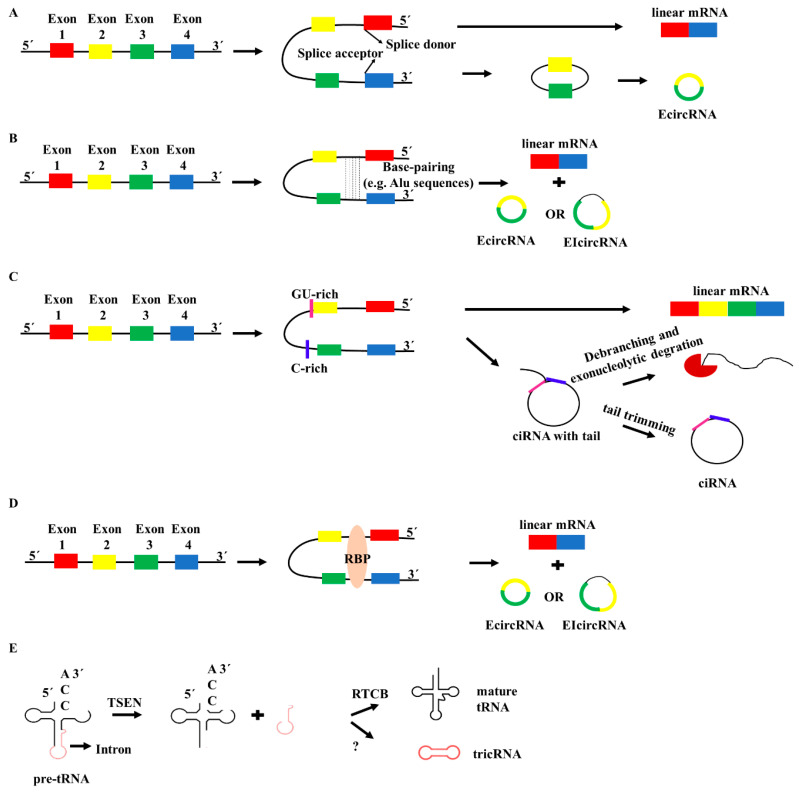
The proposed models for circular RNA (circRNA) biosynthesis. (**A**) Lariat-driven circularization or exon skipping. The pre-mRNA folds partially, encouraging the attack of the 5’ splicing site (splice donor) of the upstream intron to the 3’ splicing site (splice acceptor) of the downstream intron. This back-splicing of the folded region generates the circRNA and the rest exons generate a linear mRNA. (**B**) Intron pairing-driven circularization. Flanking reverse complementary sequences at the introns (mostly Alu sequences) mediate back-splicing generating circRNAs, i.e., EcircRNAs and EIcircRNAs. (**C**) The biogenesis of intronic circRNAs necessitates a consensus motif composed of a 7-nt GU-rich element near the 5’ splice site and an 11-nt C-rich element near the branchpoint site. (**D**) RNA-binding proteins (RBPs), such as muscleblind-like splicing regulator 1 (MBNL1) and Quaking homolog KH domain RNA-binding (QKI) protein, bring closer the donor site and the acceptor site via binding the flanks of the introns and, hence, assist circularization. (**E**) circRNAs derive from pre-tRNAs (tricRNAs), as well. tRNA splicing endonuclease (TSEN) cleaves the pre-tRNA at specific sites. The RNA 2’,3’-cyclic phosphate and 5’-OH ligase (RTCB) is essential for the ligation of the tRNA exons, but the intron ligase for the generation of the tricRNA is unknown.

**Figure 2 cancers-12-02464-f002:**
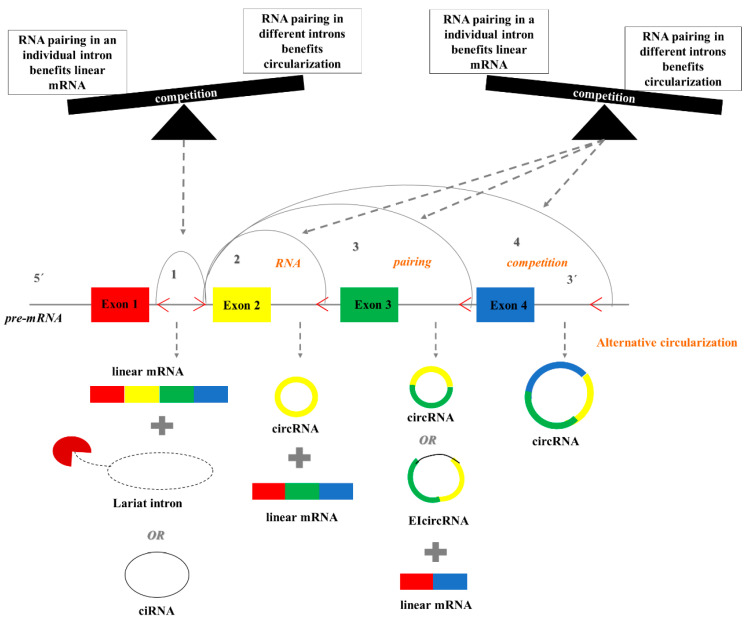
The model of alternative circularization. Multiple circRNAs and linear RNAs can be generated from a single gene locus, via RNA pairing competition. Complementary sequences within individual flanking introns favor linear mRNA formation, while complementary sequences in different flanking introns promote circularization. The competition between these reverse complementary sequences can lead to multiple circRNAs.

**Figure 3 cancers-12-02464-f003:**
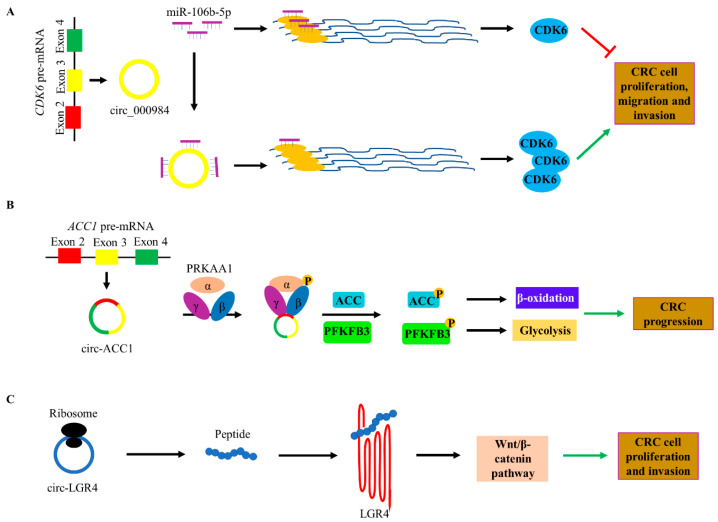
Functions of circRNAs in colorectal cancer (CRC). (**A**) Function as miRNA sponges. circ_000984 bears miRNA binding sites for miR-106b-5p. When circRNA is absent, the miRNA binds to the miRNA-response elements (MREs) of *CDK6* mRNA, leading to decreased levels of CDK6 protein and, hence, decreased proliferation, migration, and invasion. The presence of circ_000984 impedes this regulation of CDK6 expression by sequestering miR-106b-5p. (**B**) Interaction with RBPs. circACC1 interacts with β and γ subunits of PRKAA1 (AMPK), leading to its stabilization and activation. PRKAA1 phosphorylates ACC and PFKFB3, leading to increased β-oxidation and glycolysis, respectively. This mechanism promotes CRC progression. (**C**) circRNAs can encode for peptides. circ-LGR4 encodes for a peptide, which interacts with LGR4 receptor and activates Wnt/β-catenin signaling pathway, resulting in CRC cell proliferation and invasion.

**Figure 4 cancers-12-02464-f004:**
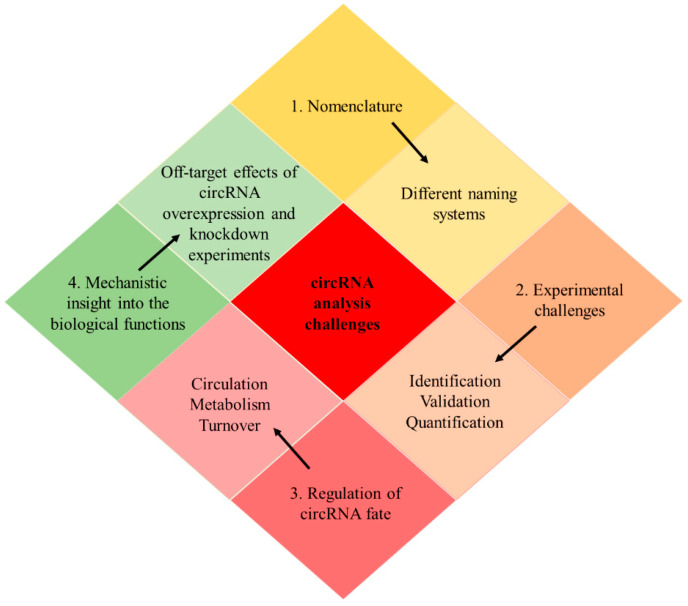
Main challenges in the field of circRNA analysis. Firstly, different nomenclature systems create ambiguity, which necessitates the establishment of a universal system, probably based on the circID of the circBase. Secondly, experimental challenges exist, which are summarized in the following categories: identification, validation, and quantification of circRNAs. Thirdly, there is a lack of information regarding the regulation of key aspects of circRNAs, such as circularization, metabolism, and turnover of these molecules. Fourthly, mechanistic insights into the biological functions of circRNAs should be gained. Novel approaches, such as the CRISPR-Cas system, could be exploited for mechanism exploration and replace the circRNA overexpression or knockdown experiments, which have off-target effects.

**Table 1 cancers-12-02464-t001:** circRNAs upregulated or downregulated in CRC, with the implication in CRC progression and metastasis.

circRNA	Gene Origin	Expression	Function	Targeted Molecules/Axes	References
circ_101555	*CSNK1G1*	Upregulation	Promotes progression	miR-597-5p/CDK6, RPA3	[53]
circ_100290		Upregulation	Promotes progression	miR-516b-5p/FZD4/Wnt/β-catenin signaling pathway	[56]
circ-ACAP2	*ACAP2*	Upregulation	Promotes proliferation, invasion, and migration	miR-21-5p/TIAM1	[57]
circ_0079993	*POLR2J4*	Upregulation	Promotes proliferation	miR-203a-3p/*CREB1*	[58]
circ_0000218	*DCLRE1C*	Upregulation	Promotes proliferation and metastasis	miR-139-3p/RAB1A	[59]
circ_0001313	*CCDC66*	Upregulation	Promotes development, progression, and metastasis	miR-510-5p/AKT2, miR-338-3p, miR-33b-5p and miR-93-5p/MYC	[60,61,62]
circ-PRMT5	*PRMT5*	Upregulation	Promotes proliferation	miR-377/E2F3/CCND1 and CDK2	[63]
circ_0071589	*FAT1*	Upregulation	Promotes tumor growth, invasion, and migration	miR-600/EZH2	[52]
circ-PIP5K1A	*PIP5K1A*	Upregulation	Promotes development	miR-1273a/JUN, IRF4, CDX2, and ZIC1	[54]
circ-DENND4C	*DENND4C*	Upregulation	Promotes proliferation, migration, and glycolysis	miR-760/SLC2A1	[50]
circ_0001900	*CAMSAP1*	Upregulation	Promotes progression	miR-328-5p/E2F1	[64]
CDR1as (ciRS-7)	*CDR1-AS*	Upregulation	Promotes progression	miR-7/EGFR and IGF1R, miR-7 independent mechanism/CMTM4 and CMTM6/PDL1	[47,49,65]
circ_0140388	*HUEW1*	Upregulation	Promotes proliferation, invasion, and migration	miR-486-5p/PLAGL2/IGF2/Wnt/β-catenin signaling pathway.	[66]
mmu_circ_003195	*NSD2*	Upregulation	Promotes metastasis	miR-199b-5p-mediated Ddr1 and Jag1 signaling	[67]
circ-LONP2	*LONP2*	Upregulation	Promotes invasion	DGCR8 & DROSHA complex in DDX1-dependent manner/pri-miR-17	[68]
circ_0007843	*ARHGAP32*	Upregulation	Promotes the invasion and migration	miR-518c-5p/*MMP2*	[69]
circ_0004680	*CCT3*	Upregulation	Promotes metastasis	miR-613/WNT3 or miR-613/VEGFA	[70]
circ-ZNF609	*ZNF609*	Upregulation	Promotes migration	miR-150-5p/GLI1/AKT	[71]
circ_000984	*CDK6*	Upregulation	Promotes cell growth and metastasis	miR-106b-5p/CDK6	[72]
circ-HIPK3	*HIPK3*	Upregulation	Promotes proliferation and metastasis	miR-1207-5p/FMNL2, miR-7/PTK2, IGF1R, EGFR, and YY1	[73,74]
circ_0055625	*DUSP2*	Upregulation	Promotes tumor growth and metastasis	miR-106b-5p/ITGB8	[75]
circ_0020397	*DOCK1*	Upregulation	Promotes cancer cell viability and invasion, and suppresses apoptosis	miR-138-5p/TERT and PD-L1	[76]
circ_0000423	*PPP1R12A*	Upregulation	Promotes cell growth and metastasis	circ-PPP1R12A-73aa/Hippo-YAP signaling pathway	[39]
circ_0053277	*NRBP1*	Upregulation	Promotes proliferation, migration, and EMT	miR-2467-3p/MMP14	[77]
circ-CDYL	*CDYL*	Downregulation	Suppresses cell growth and migration	miR-105-5p/PTEN and phosphorylation of PI3K, AKT, JAK2, and STAT5	[32]
circ_0009361	*GNB1*	Downregulation	Suppresses cell growth and metastasis	miR-582-3p/*APC2*/Wnt/β-catenin signaling pathway	[78]
circ-ITGA7	*ITGA7*	Downregulation	Suppresses proliferation and metastasis	RREB1/ITGA7/Ras signaling, miR-370-3p/NF1/Ras signaling and miR-3187-3p/ASXL1	[79,80]
circ_104916	*NEK6*	Downregulation	Suppresses migration and invasion of tumor cells by inhibiting EMT	CCNB1, CCND1	[81]
circ-SMAD7	*SMAD7*	Downregulation	Suppresses tumor metastasis by regulating EMT	CDH1 (E-cadherin), CDH2 (N-cadherin), and VIM (Vimentin)	[82]
circ-CBL.11	*CBL.11*	Downregulation	Suppresses proliferation	miR-6778-5p/YWHAE/TP53	[83]
circ-ITCH	*ITCH*	Downregulation	Suppresses development	miR-7, miR-20a-5p, and miR-214-3p/ITCH/MYC and CCND1/Wnt/β-catenin signaling pathway	[51]
circ_0021977	*PSMC3*	Downregulation	Suppresses proliferation, invasion, and migration	miR-10b-5p/CDKN1A and TP53	[84]
circ-FBXW7	*FBXW7*	Downregulation	Suppresses progression	NEK2, mTOR, and PTEN signaling pathways	[85]
circ_0014717	*CCT3*	Downregulation	Suppresses growth	CDKN2A/CDK4 and CDK6	[86]
circ_0026344	*ACVRL1*	Downregulation	Suppresses progression	miR-21-5p and miR-31-5p	[87]

**Table 2 cancers-12-02464-t002:** circRNAs upregulated or downregulated in CRC, with the implication in therapy resistance.

circRNA	Gene Origin	Expression	Impact on Therapy	Targeted Molecules/Axes	References
circ_0001313	*CCDC66*	Upregulation	Promotes resistance to radiotherapy	miR-338-3p, miR-33b-5p and miR-93-5p/MYC	[61,62]
circ_0007031	*TUBGCP3*	Upregulation	Promotes resistance to 5-FU	miR-885-3p	[92]
circ_0000504	*TUBGCP3*	Upregulation	Promotes resistance to 5-FU	miR-485-5p/STAT3	[92]
circ_0007006	*DYM*	Upregulation	Promotes resistance to 5-FU	miR-628-5p, miR-653-5p, miR-654-3p and miR-300	[92]
circ_0005963	*TMEM128*	Upregulation	Promotes resistance to oxaliplatin	miR-122-5p/*PKM2*	[95]
circ_32883	*EML5*	Upregulation	Promotes resistance to FOLFOX therapy	miR-501-5p	[94]
circ-DDX17	*DDX17*	Downregulation	Promotes sensitivity to 5-FU	miR-31-5p/*KANK1*	[93]

**Table 3 cancers-12-02464-t003:** circRNAs as potential prognostic or diagnostic biomarkers in CRC.

circRNA	Gene Origin	Biomarker Utility	References
circ_0001178	*USP25*	Diagnostic for liver metastasis	[89]
circ_0000826	*ANKRD12*	Diagnostic for liver metastasis	[89]
circ_00001666		Prognostic	[98]
circ_0122319, circ_0079480, circ_0087391	*PLOD2, ISPD, AGTPBP1*	Prognostic	[97]
CDR1as (ciRS-7)	*CDR1AS*	Prognostic	[65]
circ_0001649	*SHPRH*	Diagnostic	[99]
circ-CCDC66	*CCDC66*	Diagnostic for CEA-negative and CA19-9-negative CRC	[96]
circ-ABCC1	*ABCC1*	Diagnostic for CEA-negative and CA19-9-negative CRC	[96]
circ-STIL	*STIL*	Diagnostic for CEA-negative and CA19-9-negative CRC	[96]
circ_0026344	*ACVRL1*	Prognostic	[87]

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
