# Peer review of "Circular RNAs: A New Piece in the Colorectal Cancer Puzzle"

_cancers, 2020, doi:10.3390/cancers12092464_

Round 1
Reviewer 1 Report
This review article provides interesting information on circRNA systems that are thought to play an important role in colorectal cancer. The paper is concisely written, and provides useful information for many readers, working in the field of gastrointestinal oncology. I have no significant reservation for this manuscript. Followings are my specific comments.
- Figure 1C, linear mRNA contains exon 1 to 4 because exon seq was not contained in ciRNA.
- Figure 1C, eliminate the [+] which described upper side of ciRNA with tail.
Author Response
1. This review article provides interesting information on circRNA systems that are thought to play an important role in colorectal cancer. The paper is concisely written, and provides useful information for many readers, working in the field of gastrointestinal oncology. I have no significant reservation for this manuscript. Followings are my specific comments.
Figure 1C, linear mRNA contains exon 1 to 4 because exon seq was not contained in ciRNA.
Figure 1C, eliminate the [+] which described upper side of ciRNA with tail.
We thank the Reviewer for these comments. We modified the Figure 1C accordingly.
Reviewer 2 Report
Overall extensive review of circRNAs in cacner. Some minor queried are as follow:
- Why did the author chose CRC? is there anything unique about circRNA and CR.C? if any that can be discussed.
- Can the authors briefly describe bioinformatics tools that are available to identify circRNA ins the genome? What are the specific sequencing platforms as in Illumina,, Nanopore, etc available or protocols available?
- line 63-64: can they provide specific example of tissue specificity?
- line 83: how does the current version of hallmarks of cancer, 2011 relate to circRNA?
- line 117: can you cite the specific reference other than ref 7, which seems to be a review.
6. properly format table 2 first column
Author Response
1. Why did the author choose CRC? Is there anything unique about circRNA and CR.C? If any that can be discussed.
As it mentioned in the Introduction, CRC is characterized by an elevated mortality rate, poor prognosis of high-grade tumors, high potential of metastasis, and resistance to conventional therapies. Due to these features, researchers have focused on the identification of novel molecules and elucidation of molecular mechanisms underlying colorectal carcinogenesis. Recently, circRNAs have been studied in the context of carcinogenesis and associated with cancer initiation and development. Particularly in CRC, there are several studies that examine the expression levels and the potential role of circRNAs in this malignancy. Therefore, we decided to summarize the existing literature, in order to designate not only the current knowledge but also the existing limitations.
Introduction, lines 33-41: “Colorectal cancer (CRC) is one of the most well-studied types of human malignancies, due to its high occurrence and mortality rate worldwide.” […]
“The poor prognosis of high-grade tumors, the high potential of metastasis and the resistance to conventional therapies constitute the greatest challenges in CRC. Additionally, the absence of effective, non-invasive screening tests in the clinical practice hampers the early diagnosis of CRC. Numerous studies have focused on the elucidation of the molecular mechanisms underlying colorectal carcinogenesis, […]”
Introduction, lines 81-82: “Due to the aforementioned features and functions of circRNAs, it has been proposed that they play a pivotal role in the initiation and progression of cancer.”
Introduction, lines 90-97: “Taking all the aforementioned data into consideration, circRNAs should be scrutinized in CRC context. Until now, several studies have focused on their examination, concluding to promising results which underscore the potential role of circRNAs in CRC onset and development. Despite the great progress that has been accomplished so far, many questions remain unanswered and limitations have to be surpassed. This review aims to shed light on the current knowledge regarding the implication of circRNAs in the development of CRC, including proliferation, invasion, metastasis, and treatment resistance, and to present circRNAs which could ideally act as biomarkers and/ or therapeutic targets.”
- Can the authors briefly describe bioinformatics tools that are available to identify circRNA in the genome? What are the specific sequencing platforms as in Illumina, Nanopore, etc available or protocols available?
Taking into consideration the Reviewer’s comment, we decided to include a section (2.10. Bioinformatic tools) which briefly describes bioinformatic tools that are available to identify circRNA in the genome and important steps for library construction. We have also added the appropriate literature.
Bioinformatic tools, lines 452-466: “So far, several bioinformatic tools have been designed for the identification of circRNAs in high-throughput sequencing experiments, including CIRI, find_circ, CIRCexplorer, KNIFE, MapSplice, and circRNA_finder. The annotation of circRNAs with these tools is based on the identification of the back-splice junction, which distinguishes circRNAs from linear RNAs. For the identification based on the back-splice junction, either segmented reads or a pseudo-reference can be used. The first approach is based on splitting the sequencing reads, whereas the latter is based on a pre-defined back-splice junction and its flanking sequences in a circRNA; next, the sequencing read is directly mapped against this pseudo-reference for identification of a back-splice site [106].
However, prior to identification, library construction and sequencing are required. Several library preparation protocols can be implemented. To obtain optimal results, it is recommended that circRNA enrichment is performed prior to circRNA library construction. This can be achieved via treatment of total RNA with RNase R, an exoribonuclease which degrades linear RNA molecules, poly(A) depletion, rRNA depletion, or combination of the aforementioned strategies. circRNA sequencing reads can be either single-end or paired-end [106].”
References:
106. Chen, L.; Wang, C.; Sun, H.; Wang, J.; Liang, Y.; Wang, Y.; Wong, G. The bioinformatics toolbox for circRNA discovery and analysis. Brief Bioinform 2020, 10.1093/bib/bbaa001, doi:10.1093/bib/bbaa001.
Regarding the specific sequencing platforms (e.g. Illumina, Oxford Nanopore Technologies), etc., we omitted such information on purpose, since the platforms contain brand names. Instead of using these names, we referred to the respective high-throughput sequencing technologies, as mentioned in the following sentence:
Limitations and challenges, lines 478-482: “High-throughput RNA sequencing is used for circRNA detection, since it is a relatively unbiased procedure; however, future approaches could also adopt novel technologies for circRNA detection, such as long-read third-generation RNA sequencing, which gives information about the entire exon structure and not only the back-splice junction.”
- line 63-64: can they provide specific example of tissue specificity?
Taking into consideration the Reviewer’s comment, we included a specific example of tissue specificity, accompanied by the appropriate literature.
Introduction, lines 64-65: “For instance, CDR1as is higher expressed in murine brain tissues than in non-neural ones [5].”
References:
- Memczak, S.; Jens, M.; Elefsinioti, A.; Torti, F.; Krueger, J.; Rybak, A.; Maier, L.; Mackowiak, S.D.; Gregersen, L.H.; Munschauer, M., et al. Circular RNAs are a large class of animal RNAs with regulatory potency. Nature 2013, 495, 333-338, doi:10.1038/nature11928.
- line 83: how does the current version of hallmarks of cancer, 2011 relate to circRNA?
We thank the Reviewer for this remark. We included in the revised manuscript the two additional hallmarks in cancer (i.e. reprogramming of energy metabolism and evading immune destruction) and the respective reference (19). As it was proved by investigation in the existing literature, circRNAs affect these two characteristics. We have added the appropriate literature, as well (current references 21 and 22).
Introduction, lines 87-88: In 2011, two emerging hallmarks have been added: reprogramming of energy metabolism and evading immune destruction [19].
References:
- Hanahan, D.; Weinberg, R.A. Hallmarks of cancer: the next generation. Cell 2011, 144, 646-674, doi:10.1016/j.cell.2011.02.013.
- Zhang, L. Circular RNA: The main regulator of energy metabolic reprogramming in cancer cells. Thorac Cancer 2020, 11, 6-7, doi:10.1111/1759-7714.13251.
- Yang, L.; Fu, J.; Zhou, Y. Circular RNAs and Their Emerging Roles in Immune Regulation. Front Immunol 2018, 9, 2977, doi:10.3389/fimmu.2018.02977.
- line 117: can you cite the specific reference other than ref 7, which seems to be a review.
Prompted by the Reviewer’s comment, we added two references regarding the role of muscleblind-like splicing regulator 1 (MBNL1) and Quaking homolog KH domain RNA-binding (QKI) protein in circRNA biogenesis.
References:
- Conn, S.J.; Pillman, K.A.; Toubia, J.; Conn, V.M.; Salmanidis, M.; Phillips, C.A.; Roslan, S.; Schreiber, A.W.; Gregory, P.A.; Goodall, G.J. The RNA binding protein quaking regulates formation of circRNAs. Cell 2015, 160, 1125-1134, doi:10.1016/j.cell.2015.02.014.
- Ashwal-Fluss, R.; Meyer, M.; Pamudurti, N.R.; Ivanov, A.; Bartok, O.; Hanan, M.; Evantal, N.; Memczak, S.; Rajewsky, N.; Kadener, S. circRNA biogenesis competes with pre-mRNA splicing. Mol Cell 2014, 56, 55-66, doi:10.1016/j.molcel.2014.08.019.
- Properly format table 2 first column
We complied with the Reviewer’s comment.
Reviewer 3 Report
This review summarized the current knowledge regarding the implication of circRNAs in the development of CRC, including proliferation, invasion, metastasis, and treatment resistance, and to present circRNAs which could ideally act as biomarkers and/or therapeutic targets. The readers could understand the situations of circRNAs area well and the content of review would be very informative.
Measurement of cirRNAs may result in improvement of diagnosis for early CRC but still unclear on whether it can serve as a predictor for efficacy of chemotherapy to prolong survival in mCRC. It would be easier to understand the potential utility of cirRNAs in treatment for mCRC if this review discusses positioning between cfDNA/ctDNA and cfRNA/ctRNA.
Author Response
1. This review summarized the current knowledge regarding the implication of circRNAs in the development of CRC, including proliferation, invasion, metastasis, and treatment resistance, and to present circRNAs which could ideally act as biomarkers and/or therapeutic targets. The readers could understand the situations of circRNAs area well and the content of review would be very informative.
Measurement of cirRNAs may result in improvement of diagnosis for early CRC but still unclear on whether it can serve as a predictor for efficacy of chemotherapy to prolong survival in mCRC. It would be easier to understand the potential utility of cirRNAs in treatment for mCRC if this review discusses positioning between cfDNA/ctDNA and cfRNA/ctRNA.
We thank the Reviewer for this suggestion. Actually, we believe that the research topic of circRNAs as cancer biomarkers, with emphasis on circulating circRNAs and exosomal circRNAs, is relatively new and no pre-clinical studies evaluating their potential prognostic value alongside cfDNA/ctDNA have been carried out so far. Therefore, in our opinion, it is not so safe to speculate whether cf_circRNA/ct_circRNA is superior (or not) to cfDNA/ctDNA, regarding their value in monitoring of mCRC treatment efficacy.